# Molecular Mechanism of Male Sterility Induced by ^60^Co γ-Rays on *Plutella xylostella* (Linnaeus)

**DOI:** 10.3390/molecules28155727

**Published:** 2023-07-28

**Authors:** Shifan Li, Ke Zhang, Jiaqi Wen, Yuhao Zeng, Yukun Deng, Qiongbo Hu, Qunfang Weng

**Affiliations:** 1College of Plant Protection, South China Agricultural University, Guangzhou 510642, China; lishifan0308@163.com (S.L.); zhangke@scau.edu.cn (K.Z.); lzhonshu@163.com (J.W.); 13533061123@163.com (Y.Z.); m18825088294@163.com (Y.D.); 2Key Laboratory of Bio-Pesticide Innovation and Application of Guangdong Province, South China Agricultural University, Guangzhou 510642, China

**Keywords:** differentially expressed proteins, proteomics, pathway analysis, *Plutella xylostella*, sterile insect technique

## Abstract

*Plutella xylostella* (Linnaeus) is one of the notorious pests causing substantial loses to numerous cruciferous vegetables across many nations. The sterile insect technique (SIT) is a safe and effective pest control method, which does not pollute the environment and does not produce drug resistance. We used proteomics technology and bioinformatics analysis to investigate the molecular mechanisms responsible for the effects of different doses of radiation treatment on the reproductive ability of male *P. xylostella*. A total of 606 differentially expressed proteins (DEPs) were identified in the 200 Gy/CK group, 1843 DEPs were identified in the 400 Gy/CK group, and 2057 DEPs were identified in the 400 Gy/200 Gy group. The results showed that after 200 Gy irradiation, the testes resisted radiation damage by increasing energy supply, amino acid metabolism and transport, and protein synthesis, while transcription-related pathways were inhibited. After 400 Gy irradiation, the mitochondria and DNA in the testis tissue of *P. xylostella* were damaged, which caused cell autophagy and apoptosis, affected the normal life activities of sperm cells, and greatly weakened sperm motility and insemination ability. Meanwhile, Western blotting showed that irradiation affects tyrosine phosphorylation levels, which gradually decrease with increasing irradiation dose.

## 1. Introduction

*Plutella xylostella* (Linnaeus) prefers to feed on cruciferous vegetables of more than 40 species, including cabbage, rape and radish. It is one of the world’s worst pests of cruciferous crops [1,2]. A conservative estimate of the total global costs associated with *P. xylostella* management was reported to be USD 4 billion to USD 5 billion [3]. At present, the global control of *P. xylostella* is still dominated by chemical agents, but *P. xylostella* has a wide spectrum and rapid development of resistance, and it has developed resistance to a variety of pesticides [4,5], especially pyrethroids, carbamates, and organophosphorus insecticides [1,2], which make *P. xylostella* a highly resistant global pest [6,7,8]. *P. xylostella* has even been reported to be resistant to hard-to-resist microbial insecticides like *Bacillus thuringiensis* [9]. Thus far, *P. xylostella* resistance to *B. thuringiensis* insecticidal toxin has been reported throughout the world, in the United States, Brazil, China, Malaysia, Philippines, Japan, and other places [10,11]. The long-term use of chemical agents not only damages the ecological environment, but also suppresses populations of natural enemies. With the enhancement of people’s environmental awareness, it was of great significance to find green and safe technology for controlling *P. xylostella* [12].

The sterile insect technique (SIT) involves raising many target insects, sterilizing them by radiation treatment, and releasing a large number of sterile male insects into the field to compete with wild-type males for mating with wild-type females. Mating between sterile males and wild-type females will produce eggs that cannot hatch, resulting in a population decline [13]. SIT is an environmentally friendly pest control method compared to conventional methods. It does not produce pesticide contamination or insecticide selectivity, is not prone to resistance, and has a long-lasting control effect. After decades of development, SIT has been studied in *Glossina austeni Newstead*, *Pectinophora gossypiella* (saunders), and *Cydia pomonella* (L.), as well as other Diptera and Lepidoptera pests, with significant progress [14,15,16]. At present, the in-depth studies on radiation sterility mostly focus on the reproductive system of mammals, such as mice [17], and there are few studies on irradiated insect sperm. In recent years, with the development of molecular technology, the exploration of the internal mechanism of SIT has also been further advanced. Compared with traditional differential electrophoresis, proteomics technology based on mass spectrometry has obvious advantages in sensitivity, detection range, high throughput, accuracy, and automation [18]. Therefore, more and more research teams have begun to use proteomics technology for irradiation related research [19,20]. Although the use of SIT to control Lepidoptera pests is theoretically feasible, the mechanism of damage to the reproductive system of *P. xylostella* via irradiation and the mechanism of how the reproductive system resists different doses of irradiation are still unclear.

In this study, proteomic technology and bioinformatics analysis were used to investigate the sterility reaction mechanism of *P. xylostella* under irradiation. According to previous research results, in this study, the male pupae of *P. xylostella* were treated with group gradients of 0 Gy, 200 Gy, and 400 Gy. Proteomics was used to study the difference of protein expression levels in the testes of *P. xylostella* under different radiation doses, and the differences were mainly found from the perspective of reproduction. The relevant statistical analyses were carried out, including Gene Ontology (GO) analysis, Kyoto Encyclopedia of Genes and Genomes (KEGG) pathway analysis, subcellular localizations, and predicted functional domains, to screen the changes of differential proteins and signaling pathways in the testes of *P. xylostella* under radiation conditions and to provide a theoretical basis for further exploring the molecular mechanism of irradiation-induced male infertility of *P. xylostella*.

## 2. Results

### 2.1. Quantitative Proteomics Analysis

When the Pearson correlation coefficient is closer to −1, it is negatively correlated; when it is closer to 1, it is positively correlated. In this study, when the Pearson correlation coefficient tended to 1 within the group and −1 between the groups, it indicated good sample repeatability (*n* = 3) (Figure 1a). The mass and the length of peptides were examined to confirm the quality of the MS data. Most peptides created mass errors < 0.02 Da and included 7 to 18 amino acid residues. The findings indicated that the samples matched the requirement of the standards (Figure 1b,c).

### 2.2. Analysis of Differential Protein in P. xylostella after Irradiation

A total of 376,790 secondary profiles were obtained by mass spectroscopy. After searching the protein data database, 63,963 valid maps were obtained with a 17% utilization rate. A total of 33,496 peptides were identified by spectroscopy, of which 29,612 were specific peptides. A total of 5145 proteins were identified in this study, of which 4255 were quantifiable (quantitative proteins indicate that at least one comparison group has quantitative information) (Figure 2a). Up-regulated and down-regulated proteins were identified based on the TMT ratio. When *p*-value < 0.05, a change of differential expression greater than 1.2 was considered significantly up-regulated, and a change of differential expression less than 1/1.2 was considered significantly down-regulated. The detailed statistics of the experimental results were as follows (Figure 2b).

Venn diagram analysis was performed for the number of differentially expressed proteins (DEPs) identified in each group and the degree of overlap between them (Figure 2c). Among them, 240 proteins were differentially expressed in the three groups. A total of 333 differentially expressed proteins were found in both the 200 Gy/CK and 400 Gy/CK groups. A total of 84 proteins were only differentially expressed in the 200 Gy/CK group. A total of 245 proteins were only differentially expressed in the 400 Gy/CK group. A total of 363 proteins were only differentially expressed in the 400 Gy/200 Gy group.

### 2.3. Classification of DEPs in P. xylostella after Irradiation

In the 200 Gy/CK group, classification of the DEPs clusters was described using the following GO terms: “Signal transduction mechanisms”, “Posttranslational modification, protein turnover, chaperones”, “Intracellular trafficking, secretion, and vesicular transport”, and “Cytoskeleton” (Figure 3a). The DEPs in the biological process category were mainly grouped into “metabolic process” (158 proteins) and “cellular process” (124 proteins). The DEPs in the cell component category were mainly grouped into “cell” (77 proteins) and “organelle” (60 proteins). The DEPs in the molecular function category were mainly grouped into “binding” (257 proteins) and “catalytically active” (165 proteins) (Figure 3b). After 200 Gy irradiation, the subcellular location classification of DEPs included nucleus (175 proteins), cytoplasm (171 proteins), extracellular (94 proteins), plasma membrane (61 proteins), and mitochondria (58 proteins) (Figure 3c).

In the 400 Gy/CK group, classification of the DEPs clusters was described using the following GO terms: “Signal transduction mechanisms”, “Posttranslational modification, protein turnover, chaperones”, “Intracellular trafficking, secretion, and vesicular transport”, “Translation, ribosomal structure and biogenesis”, and “Energy production and conversion” (Figure 4a). The DEPs in the biological process category were mainly grouped into “metabolic process” (476 proteins) and “cellular process” (400 proteins). The DEPs in the cell component category were mainly grouped into “cell” (241 proteins) and “membrane” (180 proteins). The DEPs in the molecular function category were mainly grouped into “binding” (720 proteins) and “catalytically active” (508 proteins) (Figure 4b). After 400 Gy irradiation, the subcellular location classification of DEPs included cytoplasm (610 proteins), nucleus (376 proteins), extracellular (285 proteins), mitochondria (218 proteins), and plasma membrane (188 proteins) (Figure 4c).

In the 400 Gy/200 Gy group, classification of more DEPs clusters was described using the following GO terms: “Posttranslational modification, protein turnover, chaperones”, “Signal transduction mechanisms”, “Intracellular trafficking, secretion, and vesicular transport”, “Translation, ribosomal structure and biogenesis”, “Energy production and conversion”, “Carbohydrate transport and metabolism”, and “Cytoskeleton” (Figure 5a). After 400 Gy irradiation, compared to 200 Gy irradiation, the DEPs in the biological process category were mainly grouped into “metabolic process” (545 proteins) and “cellular process” (461 proteins). The DEPs in the cell component category were mainly grouped into “cell” (286 proteins) and “membrane” (196 proteins). The DEPs in the molecular function category were mainly grouped into “binding” (830 proteins) and “catalytically active” (567 proteins) (Figure 5b). After 400 Gy irradiation, the subcellular location classification of DEPs included cytoplasm (682 proteins), nucleus (430 proteins), extracellular (309 proteins), mitochondria (236 proteins), and plasma membrane (201 proteins) (Figure 5c).

### 2.4. Enrichment Analysis of DEPs in P. xylostella after Irradiation

Further KEGG enrichment analysis was performed on the function of DEPs in order to obtain potential signaling pathways in which DEPs may be involved. The DEPs in the 200 Gy/CK group were involved in 40 relevant pathways, the DEPs in the 400 Gy/CK group were associated with 36 pathways, and the DEPs in the 400 Gy/200 Gy group were associated with 38 pathways. Some important KEGG pathways enriched for DEPs in the three groups are shown in Figure 6.

In the 200 Gy/CK group, up-regulated proteins were mainly enriched in pxy00190 oxidative phosphorylation (*p* = 0.0124495314091878) (Figure 7), pxy00910 nitrogen metabolism (*p* = 0.0127166975945574), pxy00270 Cysteine and methionine metabolism (*p* = 0.0432979319307483), and pxy00220 Arginine biosynthesis (*p* = 0.0046872098346179). The down-regulated proteins were mainly enriched in transcription related pathways (Figure 8), such as pxy03013 RNA transport (*p* = 0.0456349122598015), pxy03040 spliceosome (*p* = 0.00596143898975832), and pxy03022 basal transcription factors (*p* = 0.0219739068054894).

In the 400 Gy/CK group, up-regulated proteins were mainly enriched in pxy00052 galactose metabolism (*p* = 0.00330912713848497) (Figure 9), pxy03460 Fanconi anemia pathway (*p* = 0.00364682682546935), and pxy00592 alpha-linolenic acid metabolism (*p* = 0.0128620802533995). The down-regulated proteins were mainly enriched in pxy03010 ribosome (*p* = 0.0124654621445671) (Figure 10).

In the 400 Gy/200 Gy group, up-regulated proteins were mainly enriched in pxy03008 Ribosome biogenesis in eukaryotes (*p* = 0.012489820154116), pxy04130 SNARE interactions in vesicular transport (*p* = 0.0377266515337367), pxy04137 mitopha Gy-animal (*p* = 0.0434755479007902), and pxy03460 Fanconi anemia pathway (*p* = 0.0443262343121452) (Figure 11). The down-regulated proteins were mainly enriched in pxy04150 mTOR signaling pathway (*p* = 0.0424376647603529) (Figure 12), pxy00515 mannose type O-glycan biosynthesis (*p* = 0.0439169920867376), pxy04145 phagosome (*p* = 0.0256823815552166), pxy00250 alanine, aspartate, and glutamate metabolism (*p* = 0.00526714963588115), and pxy00220 Arginine biosynthesis (*p* = 0.01474651626778).

### 2.5. Functional Enrichment

In the 200 Gy/CK group, functional enrichment with GO analysis revealed the significant relationships of the differentially up-regulated proteins with protein functions related (all *p* < 0.01) to proteolysis, tidase activity, aminopeptidase activity, exopeptidase activity, metallopeptidase activity, metalloexopeptidase activity, glutathione synthase activity, cytoplasmic dynein complex, membrane part, and mitochondrial respiratory chain (Figure 13a–d). Moreover, functional enrichment with GO analysis revealed the significant relationships of the differentially down-regulated proteins with protein functions related (all *p* < 0.01) to regulation of RNA metabolic process, regulation of macromolecule biosynthetic process, regulation of cellular biosynthetic process, protein heterodimerization activity, and lysozyme activity (Figure 13e,f).

In the 400 Gy/CK group, functional enrichment with GO analysis revealed the significant relationships of the differentially down-regulated proteins with protein functions related (all *p* < 0.01) to transaminase activity, peptidoglycan muralytic activity, structural molecule activity, structural constituent of cytoskeleton, structural constituent of ribosome, single-stranded DNA binding, supramolecular complex, supramolecular polymer, supramolecular fiber, polymeric cytoskeletal fiber, and microtubule (Figure 14a–d).

In the 400 Gy/200 Gy group, functional enrichment with GO analysis revealed the significant relationships of the differentially up-regulated proteins with protein functions related (all *p* < 0.01) to the nucleus and the spliceosome complex (Figure 15a). Moreover, functional enrichment with GO analysis revealed the significant relationships of the differentially down-regulated proteins with protein functions related (all *p* < 0.01) to single-stranded DNA binding, transaminase activity, structural molecule activity, structural constituent of cytoskeleton, guanyl nucleotide binding, guanyl ribonucleotide binding, GTP binding, supramolecular complex, supramolecular polymer, supramolecular fiber, polymeric cytoskeletal fiber, microtubule organizing center, microtubule cytoskeleton, microtubule, microtubule cytoskeleton organization, and microtubule polymerization or depolymerization (Figure 15b–e).

### 2.6. Western Blot Analysis

Protein-specific expression levels of tyrosine phosphorylated proteins in *P. xylostella* testis after treatment with different doses of irradiation were analyzed by Western blotting. As shown in Figure 16, the number of proton bands decreases and the color becomes lighter as the irradiation group increases, indicating a decreasing trend in the expression levels of tyrosine phosphorylated proteins in the testes of *P. xylostella*.

## 3. Discussion

Studies of SIT are relatively common in Diptera, Hemiptera, and Lepidoptera species, but most of the studies of Lepidoptera species under SIT projects still focus on the screening of sterile dose of various pests and the detection of reproductive ability of insects by irradiation [21,22]. However, the molecular mechanism of irradiation-induced sterility is still unclear due to the lack of studies of DEPs in insect sperm cells after irradiation. In this study, TMT-labeled quantitative proteomics was used to investigate the testicular tissue of *P. xylostella* after irradiation, and the related proteomic changes in the testes of the male *P. xylostella* were identified by analyzing the DEPs, which provided the possibility for further study on the molecular mechanism of irradiated infertility of *P. xylostella.*

### 3.1. Effect of 200 Gy/CK Group on the Spermial Tissue of P. xylostella

Most of the energy required for cellular life activities is provided by ATP produced by mitochondria during aerobic respiration, and oxidative phosphorylation is an important stage of cellular aerobic respiration. The TCA cycle is a hub for carbohydrate, lipid, and protein communication and conversion. However, in general, irradiation can lead to the inhibition of energy metabolism and amino acid metabolism, damage the respiratory chain, hinder the growth of cells, affect the normal life activities of cells [23], and even cause abnormal energy metabolism and induce cell apoptosis [24]. After 200 Gy irradiation, the oxidative phosphorylation pathway and the TCA cycle pathway of *P. xylostella* were significantly enhanced (all *p* < 0.05), and a variety of pathways related to amino acid metabolism were also up-regulated. It was speculated that under the stress of 200 Gy irradiation, the testis cells of *P. xylostella* can obtain more energy by enhancing energy and amino acid metabolism and accelerate protein synthesis, thus promoting cell repair and helping the body resist the radiation damage.

Regulation of eukaryotic gene expression can take place at multiple levels, with post-transcriptional regulation being particularly important for motor tissues. Compared to other tissues, spermatozoa have the highest transcriptional activity and variable splicing frequencies. The spliceosome complex is composed of five kinds of small nuclear RNA (snRNA) in the nucleus combined with proteins. The correct assembly of snRNA was an essential prerequisite for the formation of RNA splicing complexes. Abnormal regulation of the spliceosome and accumulation of erroneous RNA can lead to tissue and organ dysfunction and many diseases [25,26]. After 200 Gy irradiation, the spliceosome pathway was significantly decreased (*p* < 0.05), and SNRP70, P62, SF3a, SF3b, SPF45, and Brr2 in the splicing-complexes U1, U2, and U5 were significantly down-regulated (all *p* < 0.05).

RNA transport from the nucleus to the cytoplasm is the basis of translational expression. After 200 Gy irradiation, the expressions of NUP205, NUP155, Gemin5, EIF4E, and PAIP1 proteins in the testes of *P. xylostella* were down-regulated (all *p* < 0.05). NUP155 and NUP205 are a key component of the nuclear pore complex (NPC). The NPC is a 100 MDa macro-molecular assembly that spans the nuclear envelope and mediates molecular exchange between the cytoplasm and nucleoplasm [27,28]. Gemin5 is not only an RNA-binding protein for motoneuron complex survival, but also for translational regulation and ribosome binding [29,30]. Changes in the level and activity of the EIF4E complex determine the binding rate of ribosomes to mRNA, which was the rate-limiting point for translation initiation [31]. PAIP1 regulates mRNA translation efficiency and development, and influences the cell cycle by interacting with several proteins [32,33,34]. The expression of these proteins enriched in the RNA transport pathway was significantly down-regulated (*p* < 0.05), reducing the efficiency of RNA transport in the sperm cells of the *P. xylostella*, thus affecting the subsequent gene expression.

At the same time, SAP18, MAGOH, PAP, and FIP1 proteins in the RNA monitoring pathway of *P. xylostella* spermatozoa were also significantly decreased (all *p* < 0.05). SAP18 and MAGOH proteins are components of splice-dependent multiprotein exon-junction complexes [35,36]. Precursor RNA (pre-mRNA) is polyadenylated by enzymes called poly(A) polymerases (PAPs), which function in a 3′-end processing complex containing a large number of proteins components [37]. FIP1 can bind RNA and stimulate PAP activity [38]. Studies have shown that both favorable and unfavorable environmental factors can affect the physiological and behavioral phenotypes of offspring through non-coding RNA-mediated transgenerational inheritance [39]. Combined with the above changes, we speculated that after 200 Gy irradiation, normal RNA transcription, splicing, and transport processes were impeded, resulting in an increase of defective RNA that severely affects the expression of subsequent genes and may be passed on to offspring.

### 3.2. Effect of 400 Gy/CK Group on the Spermial Tissue of P. xylostella

Under normal physiological conditions, most cells rarely rely on glycolysis for energy. However, 400 Gy irradiation induced the enhancement of glycolytic pathway and the inhibition of the oxidative phosphorylation pathway in the sperm, which is similar to the Warburg effect [40]. Once the most essential component of the ribosome, ribosomal proteins play an important role in the ribosome’s translation process. Its abnormal regulation can lead to cell cycle arrest and even apoptosis [41,42]. After 400 Gy irradiation, fifteen 60S Ribosome large subunit proteins, ten 40S small ribosome large subunit proteins, and two 39S mitochondrial ribosome large subunit proteins belonging to the “pxy03010 Ribosome” pathway were significantly down-regulated (all *p* < 0.05). This also indicates that irradiation may cause damage to the ribosome function of cell and thus hinder protein synthesis in the cell.

Mitochondria provide energy for cells, play a regulatory role in cell apoptosis, and are also involved in the production of oxygen free radicals and the removal of misfolded proteins, playing an important role in maintaining cell physiological functions. When cells encounter external stimuli, they need to form autophagosomes, an alternative death mechanism to apoptosis [43]. After 400 Gy irradiation, the expression levels of ATG7, ATG8, ATG9, ATG16, and ATG18 proteins in the autophagy pathway were significantly up-regulated (all *p* < 0.05), and these proteins are all important autophagy related proteins [44,45]. In the apoptotic pathway, caspase-8 and APAF-1 proteins were significantly up-regulated (all *p* < 0.05). Caspase-8 is the initiating enzyme of the apoptosis pathway, which can be activated by a series of caspase cascades to perform the apoptotic process [46]. The APAF-1 protein acts as a bridge in this process and is central to the entire apoptotic complex [47].

Under normal conditions, the genome of an organism is stable, but stimuli from external or internal conditions can cause DNA damage. Germ cells are highly sensitive to DNA damage due to their frequent long-term fission reproduction and active DNA replication [48], which is passed on to the next generation [49]. The Fanconi anemia pathway and homologous recombination pathway are important pathways for DNA damage repair. The Fanconi anemia pathway is also closely related to germ cell development [50]. After 400 Gy irradiation, the Fanconi anemia pathways and homologous recombination pathways related to DNA damage repair were significantly enriched and up-regulated (*p* < 0.05) in *P. xylostella* sperm cells. This indicates that the DNA of the sperm tissue cells was damaged to some extent. These results are consistent with our previous findings that the cells in the testes were significantly apoptotic and the nucleus chromatin shrunk after 400 Gy irradiation [51].

Based on the above analysis, we speculate that the cause of sterility in *P. xylostella* after 400 Gy irradiation may be due to severe damage to the ribosome, mitochondria, and DNA of the testis tissue, created by amino acid metabolism, blocking protein synthesis, energy metabolism, cell growth, and other normal physiological processes. It has severe effects on sperm motility and fertilization, and even directly induces the apoptosis of spermatozoa tissue cells, resulting in male sterility.

### 3.3. Effect of 400 Gy/200 Gy Group on the Spermial Tissue of P. xylostella

The effects of different irradiation groups (400 Gy/200 Gy) on the sperm of *P. xylostella* were basically the same as those of the control group (200 Gy/CK, 400 Gy/CK), but the changes of the 400 Gy/200 Gy irradiation group were more obvious in the following three aspects. The Wnt signaling pathway was significantly up-regulated (*p* < 0.05), the mTOR signaling pathway was inhibited, and the tyrosine phosphorylation level was decreased. The classical Wnt signaling pathway is closely related to cell development and differentiation and is important for the normal function of the adult testis [52]. Studies have found that the Wnt signaling pathway is usually activated or up-regulated to accelerate the induction of apoptosis [53,54]. mTOR is a serine/threonine protein kinase involved in the regulation of cell growth, apoptosis, autophagy, and other processes [55]. mTOR gene expression is positively correlated with sperm motility [56]. Conditional knockout of the mTOR gene in mouse testicular cells results in decreased sperm number, increased sperm malformation rate, and increased germ cell apoptosis in mouse testicular tissue [57]. An increasing number of post-translational phosphorylated tyrosine proteins have been shown to be involved in many biological processes, including sperm productivity, capacitation and acrosome reactions [58,59,60]. Asthenospermia is associated with decreased protein tyrosine phosphorylation [61]. In this study, the levels of protein tyrosine phosphorylation in the spermatozoa of *P. xylostella* decreased with increasing irradiation doses. We speculate that the decrease in tyrosine phosphorylation may adversely affect sperm motility, and thus ultimately fertilization.

## 4. Materials and Methods

### 4.1. Rearing and Irradiation Treatment of P. xylostella

The pupae of *P. xylostella* were collected from a cabbage mustard field in Guangdong Province, China. They were reared and maintained under laboratory conditions at 25 ± 1 °C 60–70% RH, and L:D = 8 h:16 h. The pupae were confined in a cage (50 cm in length, 45 cm in width, 45 cm in height) with 10% honey for feeding and allowed to mate. Male pupae were collected for irradiation treatment when the surface was brown and patterned. The mature pupae that grew to the sixth day were exposed to ^60^Co γ-rays irradiation. Six-day-old male pupae (due to radiation resistant and convenient for in vitro experiments) with the same growth pattern were selected with tweezers, placed in a petri dish with a diameter of 9 cm, and irradiated with ^60^Co γ-rays in groups of 0 Gy, 200 Gy and 400 Gy; the group rate was 16.67 Gy/min, and 0 Gy was set as the blank control. Three replicates were set for each treatment of 0 Gy, 200 Gy, and 400 Gy, and 200 male pupae were collected for each replicate.

### 4.2. Anatomy of P. xylostella Testis and Extraction of Testis Protein

Three replicates were set for each treatment of 0 Gy, 200 Gy, and 400 Gy, and 200 male pupae were collected for each replicate. Each group of irradiated males were selected for dissection 24 h after emergence. Each adult to be dissected was immersed in alcohol (100%) for 3–5 s to soak the epidermis and kill them. It was then placed in PBS buffer (pH = 7.4) to rinse off the alcohol. The *P. xylostella* was placed on a glass slide with a drop 10 μL of PBS buffer, then placed under a stereoscope (Motic SMZ-171). The abdominal epidermis was gently peeled off with a dissecting needle; the testes were taken out and immediately put into PBS buffer. The testis tissue samples were temporarily stored in an ice box, and then the testis tissue samples were added to 4 volumes of lysis buffer (8 M urea, 1% protease inhibitor, 3 μM TSA, 50 mM NAM, and 2 mM EDTA) and lysed by sonication. The sample was then centrifuged at 12,000 rpm at 4 °C for 10 min, the cell fragments were removed, and the supernatant was transferred to a new centrifuge tube for testing.

### 4.3. Protein Concentration Determination

Referring to the Bradford method (1976), using the principle that Coomassie brilliant blue G250 reacts with aromatic amino acids in proteins, a protein standard curve was prepared in a 96-well plate [62]. The protein concentration was calculated by measuring the absorbance at a wavelength of 562 nm with a microplate reader.

Preparation of the standard curve: First, 10 mg of bovine serum albumin was weighed and a PBS buffer of 10 mL was added to prepare a standard solution of bovine serum albumin at a concentration of 10 mg/mL, which was used to make the protein standard curve. The samples shown in Table 1 were loaded into a 96-well plate, mixed evenly, and repeated three times. After capping, the 96-well plate with the samples was placed at room temperature for 2 min, and then the OD value was measured at 562 nm (shaking first), with the content of bovine serum albumin (μg) as the abscissa and the OD value as the ordinate; a standard curve was then plotted. The standard curve equation for proteins determined in this paper was y = 1.145x + 0.3857, and R^2^ was 0.9861.

According to the same processing method, a certain gradient concentration of protein extract was pipetted into a 96-well plate, 200 μL of Coomassie brilliant blue G-250 was added, the sample was well mixed, and its OD_562_ was measured with a microplate reader.

### 4.4. SDS-PAGE Electrophoresis Detection

There were three replicates for each treatment of 0 Gy, 200 Gy, and 400 Gy, for a total of nine samples. A 12% polyacrylamide gel was used; the sample was loaded and the electrophoresis was run at 80 V for 30 min. Then, electrophoresis at a constant voltage of 120 V was run until bromophenol blue simply ran out of the separation gel. After electrophoresis was completed, the samples were put into Coomassie brilliant blue R-250 dye solution for 2 h, and shaken continuously during this period.

### 4.5. Trypsin Digestion and TMT Labeling

For digestion, the protein solution was reduced with 5 mM dithiothreitol for 30 min at 56 °C and alkylated with 11 mM iodoacetamide for 15 min at room temperature in darkness. The protein samples were then diluted by adding 100 mM TEAB to a urea concentration less than 2 M. Finally, trypsin was added at a 1:50 trypsin-to-protein mass ratio for the first overnight digestion and at a 1:100 trypsin-to-protein mass ratio for the second 4 h digestion.

After trypsin digestion, peptide was desalted by Strata X C18 SPE column (Phenomenex) and vacuum dried. The peptides were reconstituted in 0.5 M TEAB and processed according to the manufacturer’s protocol for the TMT kit. Temporarily, a unit of TMT reagent was thawed and reconstituted in acetonitrile. The peptide mixtures were then incubated for 2 h at room temperature and pooled, desalted, and dried by vacuum centrifugation.

### 4.6. High Performance Liquid Chromatography (HPLC) Fractionation

The tryptic peptides were fractionated into fractions by high pH reverse-phase HPLC using a Thermo Scientific BetaSil C18 column (5 μm particles, 10 mm ID, 250 mm length). Briefly, peptides were first separated with a gradient of 8% to 32% acetonitrile (pH 9.0) over 60 min into 60 fractions. The peptides were then combined into 6 fractions and dried by vacuum centrifugation.

### 4.7. Mass Spectrum (MS) Analysis

The peptides were dissolved in liquid chromatography mobile phase A (0.1% (*v*/*v*) formic acid in water) and separated using an EASY-nLC 1000 ultra-high performance liquid chromatography system. Mobile phase A (aqueous phase): 0.1% formaldehyde, 2% ethanol; mobile phase B (organic phase): 0.1% formic acid, 90% acetonitrile. Liquid gradient settings: 0–60 min, 6–22% B; 60–75 min, 22–30% B; 75–77 min, 30–80% B; 77–80 min, 80% B. The flow rate was maintained at 0.5 μL/min.

### 4.8. Database Search

The resulting MS/MS data were processed using MaxQuant search engine (v.1.5.2.8). Tandem mass spectra were searched against the human uniport database concatenated with the reverse decoy database. Trypsin/P was specified as a cleavage enzyme, allowing for up to 4 missing cleavages. The mass tolerance for the precursor ions was set to 20 ppm in the first search and 5 ppm in the main search, and 0.02 Da for fragment ions. Carbamidomethylation on Cys was a fixed modification, while acetylation modification and oxidation on Met were specified as variable modifications. FDR was adjusted to <1% and minimum score for modified peptides was set >40.

### 4.9. Bioinformatics Analysis

A bioinformatics analysis was performed on the quantitative data of the proteome and the data of the DEPs in order to find the variation laws of the DEPs. Bioinformatics analyses included gene ontology (GO) analysis, KEGG pathway analysis, protein domain analysis, and subcellular localization analysis. This paper used the bioinformatics analyses of the related software or websites as shown in Table 2.

### 4.10. Western Blot

Three replicates were performed for each treatment of 0 Gy, 200 Gy, and 400 Gy, for a total of nine samples. Sample buffer (4×) was added to each sample (30 μg), further diluted to 1×, and the protein was completely denatured by heating in boiling water for 10 min. The protein samples were subjected to electrophoresis, electroporation, and blocking, then the primary anti-phosphotyrosine antibody (Jingjie PTM Biolab) was added and incubated at 4 °C overnight. After washing the membrane, the secondary anti-goat anti-mouse IgG (Thermos) was added and incubated at room temperature for 2 h, and the substrate was exposed.

## 5. Conclusions

Male pupae of *P. xylostella* responded to different irradiation doses by different mechanisms. After 200 Gy irradiation, spermatophore cells of *P. xylostella* resisted radiation damage by accelerating metabolism, increasing energy supply, amino acid metabolism, protein synthesis, and other ways. However, the transcription-related pathways were inhibited, which may hinder the normal progress of RNA transcription, splicing, transport, and other processes, resulting in the increase of defective RNA which may be inherited to future generations. Compared with 200 Gy irradiation group, 400 Gy irradiation can promote the expression of signaling pathways related to proteolysis and gene expression in tissue cells and inhibit the expression of signaling pathways related to protein synthesis and cytoskeleton synthesis. Moreover, irradiation influences the tyrosine phosphorylation level, which decreases with the increase of the irradiation dose.

## Figures and Tables

**Figure 1 molecules-28-05727-f001:**
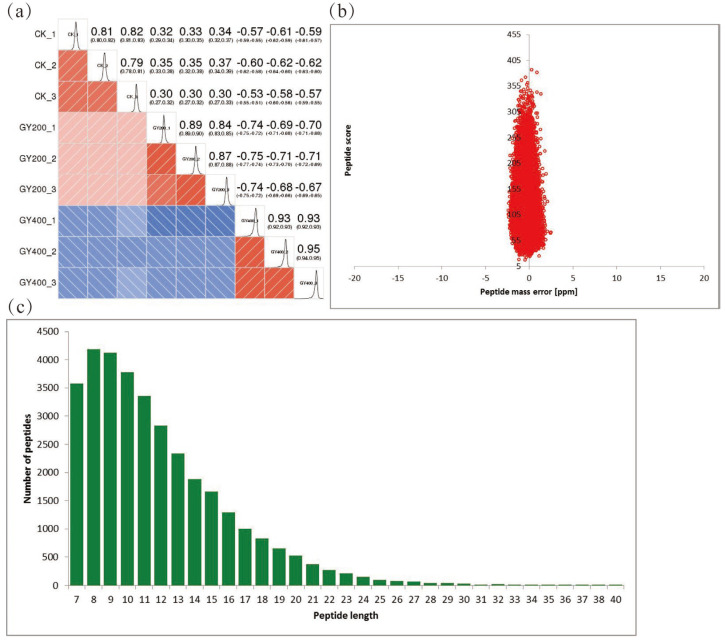
Quality validation of the proteomic data. (**a**) Pearson’s correlation coefficient of protein quantitation. (**b**) Length distribution of all identified peptides. (**c**) Mass distribution of all identified peptides.

**Figure 2 molecules-28-05727-f002:**
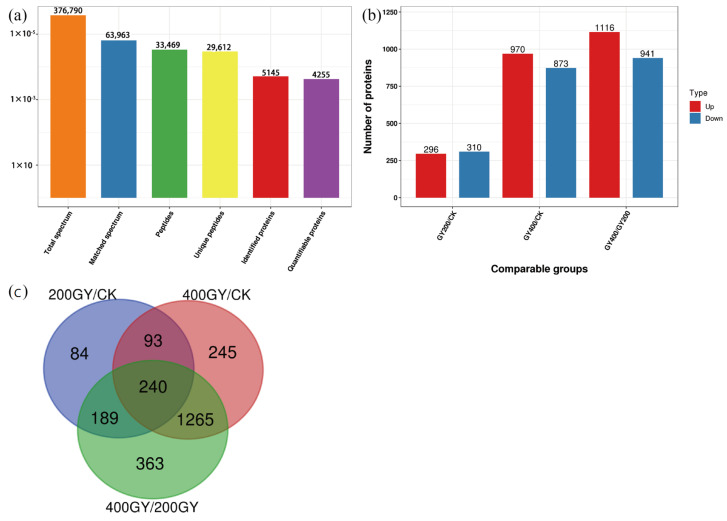
Classification and annotation of the DEPs. (**a**) Results of liquid chromatography-tandem mass spectrometry analysis of the proteins. (**b**) Number of DEPs. (**c**) Venn diagram of differential protein expression.

**Figure 3 molecules-28-05727-f003:**
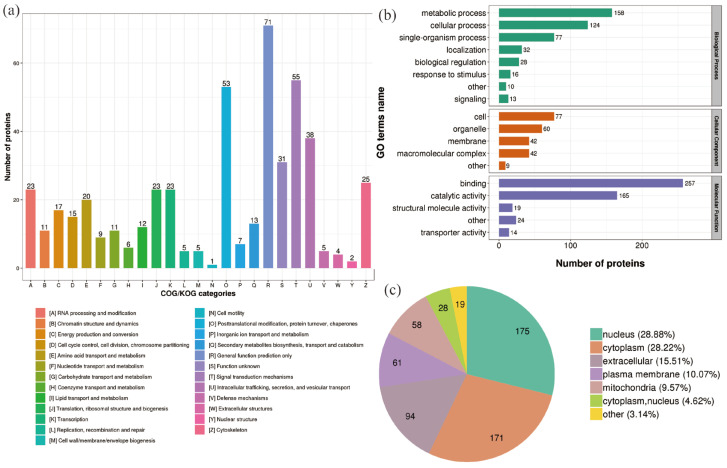
Classification of DEPs (200 Gy/CK group). (**a**) COG/KOG category analysis of DEPs. (**b**) GO analysis of DEPs. (**c**) Predicted subcellular localization of DEPs.

**Figure 4 molecules-28-05727-f004:**
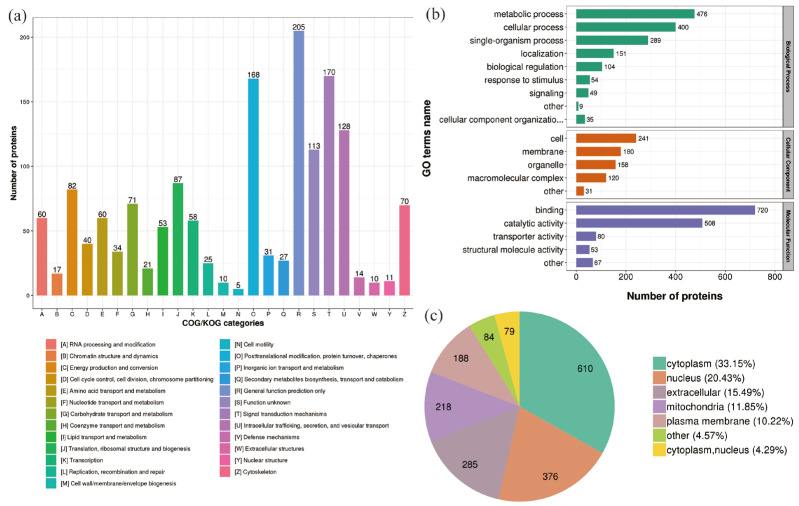
Classification of DEPs (400 Gy/CK group). (**a**) COG/KOG categories analysis of DEPs. (**b**) GO analysis of DEPs. (**c**) Predicted subcellular localization of DEPs.

**Figure 5 molecules-28-05727-f005:**
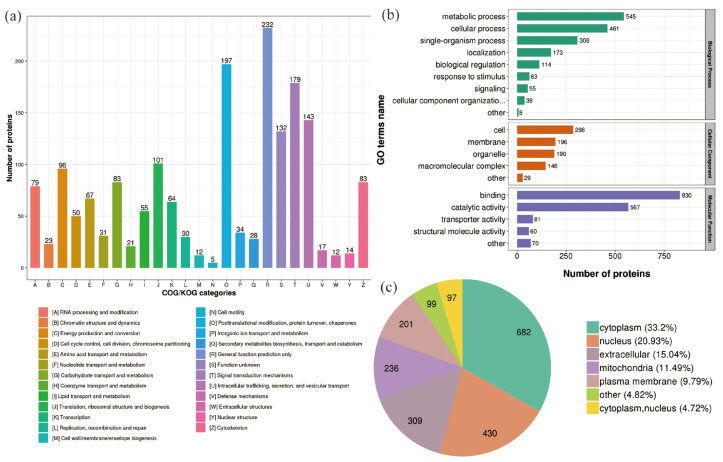
Classification of DEPs (400 Gy/200 Gy group). (**a**) COG/KOG categories analysis of DEPs. (**b**) GO analysis of DEPs. (**c**) Predicted subcellular localization of DEPs.

**Figure 6 molecules-28-05727-f006:**
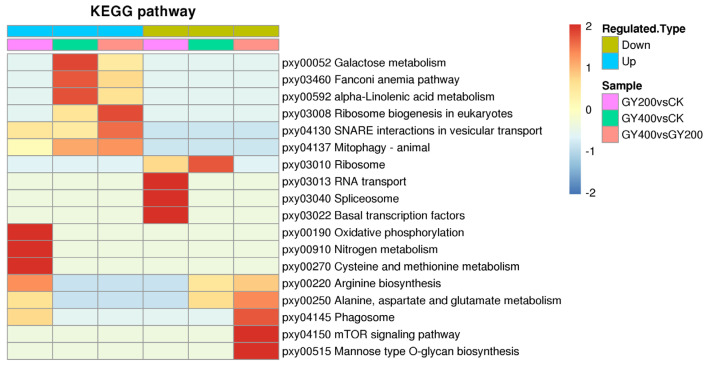
KEGG enrichment analysis of DEPs.

**Figure 7 molecules-28-05727-f007:**
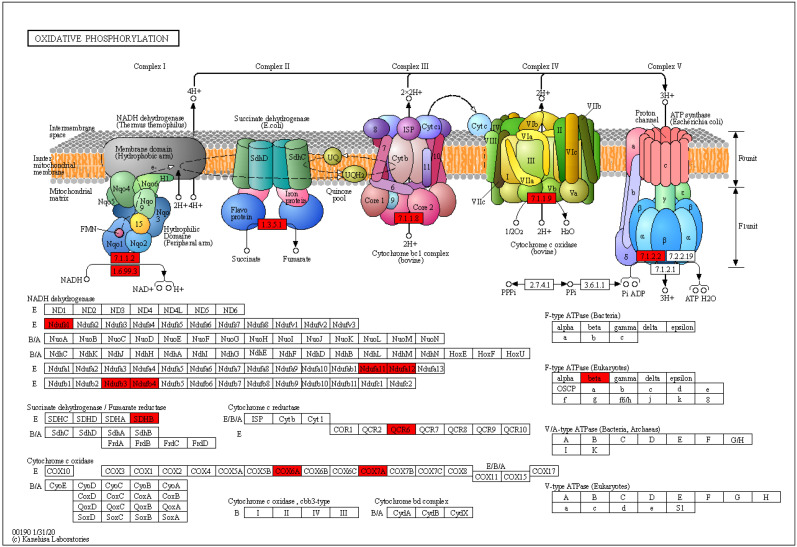
Oxidative phosphorylation pathway in the seminal tissues of *P. xylostella* in the 200 Gy/CK group. Red indicates differentially up-regulated proteins; green indicates differentially down-regulated proteins.

**Figure 8 molecules-28-05727-f008:**
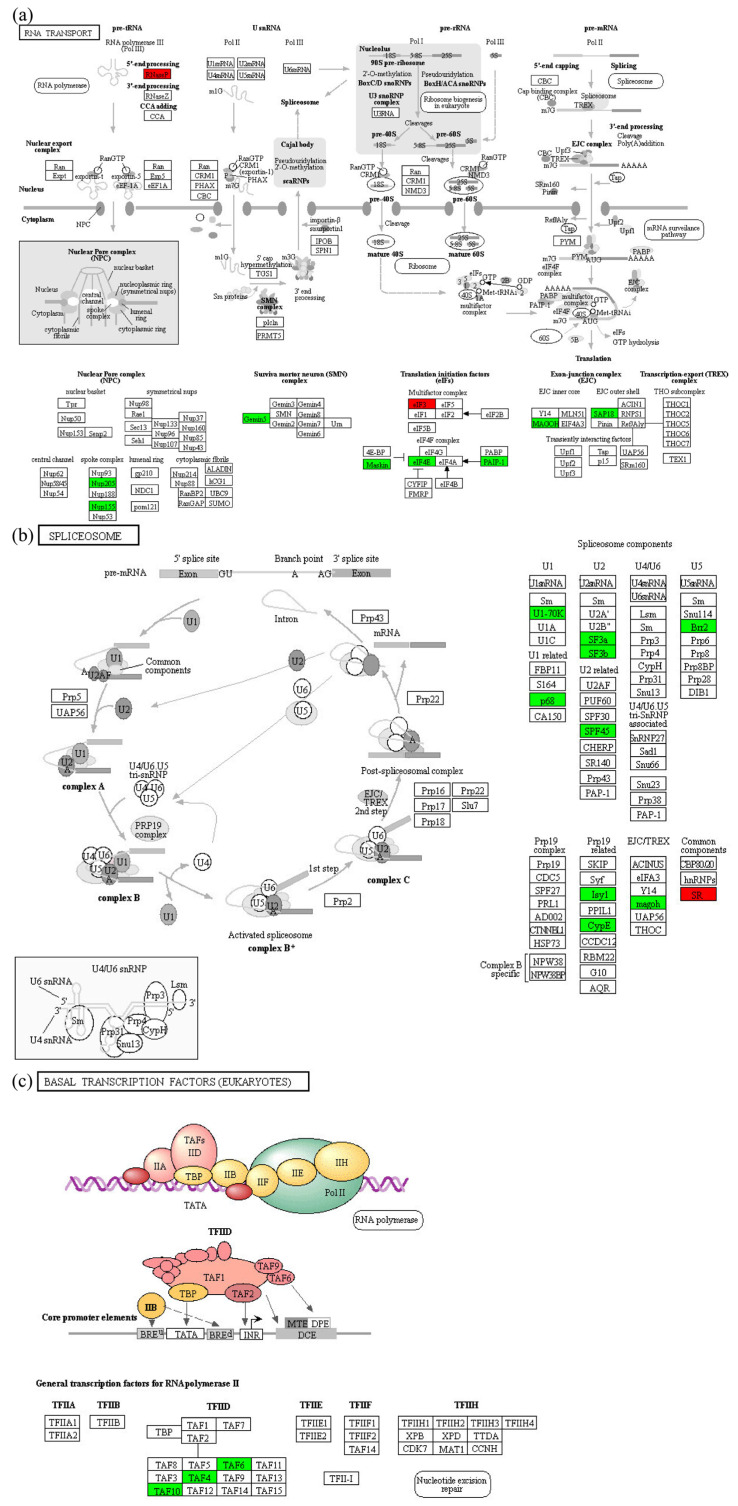
Transcription related pathways in the seminal tissues of *P. xylostella* in the 200 Gy/CK group. (**a**) RNA transport. (**b**) Spliceosome. (**c**) Basal transcription factors. Red indicates differentially up-regulated proteins; green indicates differentially down-regulated proteins.

**Figure 9 molecules-28-05727-f009:**
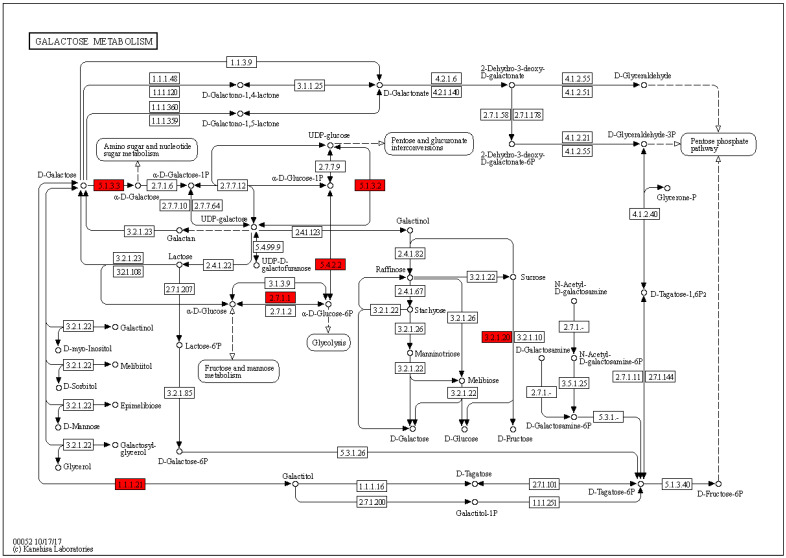
Galactose metabolism in the seminal tissues of *P. xylostella* in the 400 Gy/CK group. Red indicates differentially up-regulated proteins.

**Figure 10 molecules-28-05727-f010:**
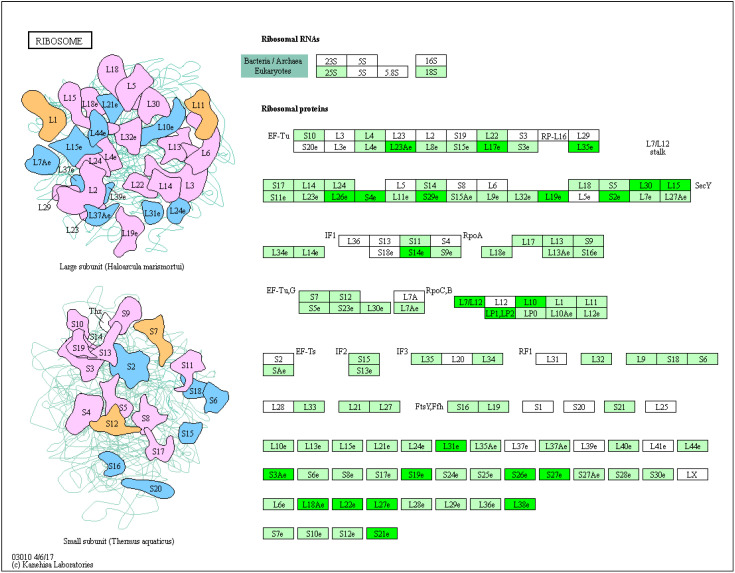
Ribosome in the seminal tissues of *P. xylostella* in the 400 Gy/CK group. Green indicates differentially down-regulated proteins.

**Figure 11 molecules-28-05727-f011:**
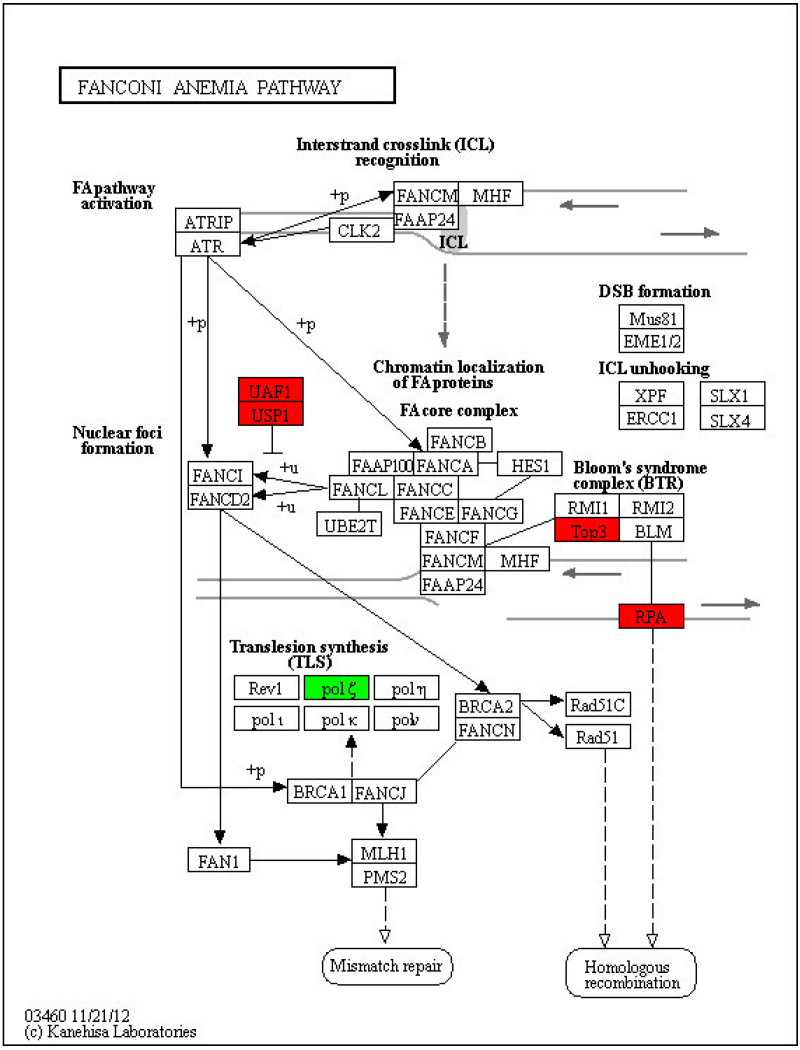
Fanconi anemia pathway of *P. xylostella* in the 400 Gy/200 Gy group. Red indicates differentially up-regulated proteins; green indicates differentially down-regulated proteins.

**Figure 12 molecules-28-05727-f012:**
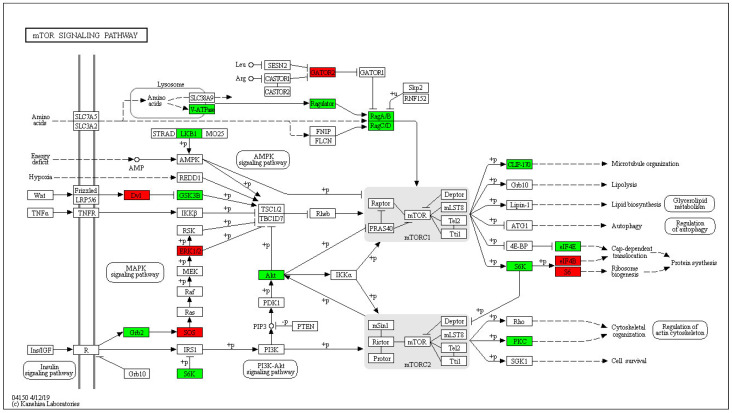
mTOR signaling pathway of *P. xylostella* in the 400 Gy/200 Gy group. Red indicates differentially up-regulated proteins; green indicates differentially down-regulated proteins.

**Figure 13 molecules-28-05727-f013:**
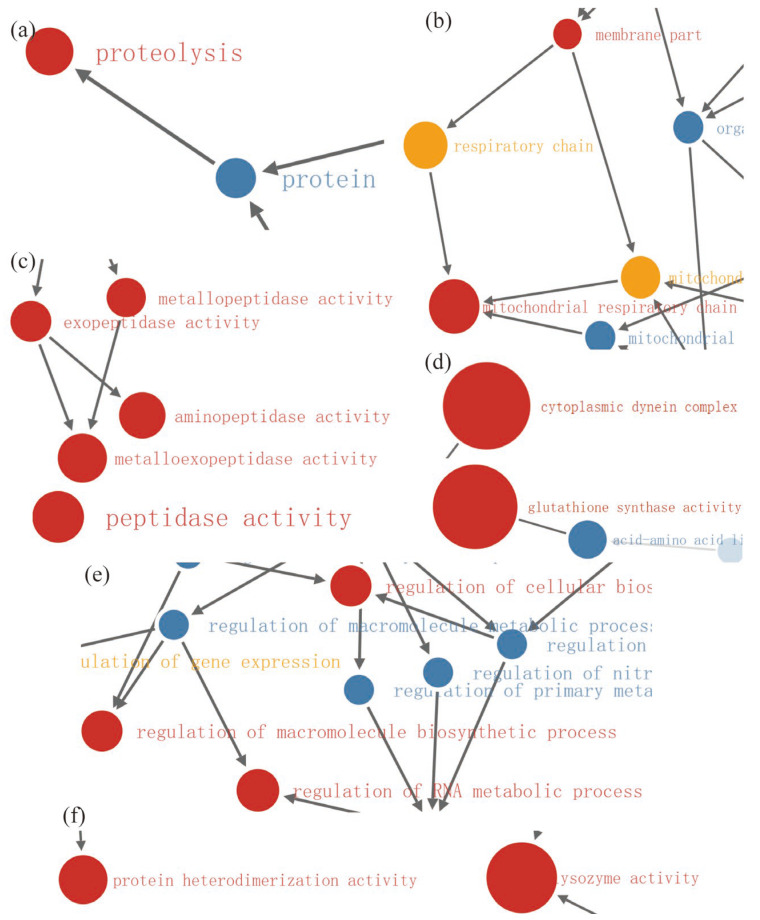
Functional enrichment diagram of 200 Gy/CK group. (**a**–**d**) Functional enrichment diagram of up-regulated proteins. (**e**,**f**) Functional enrichment diagram of down-regulated proteins. Red indicates *p* < 0.01; yellow indicates *p* < 0.05; blue indicates no significance.

**Figure 14 molecules-28-05727-f014:**
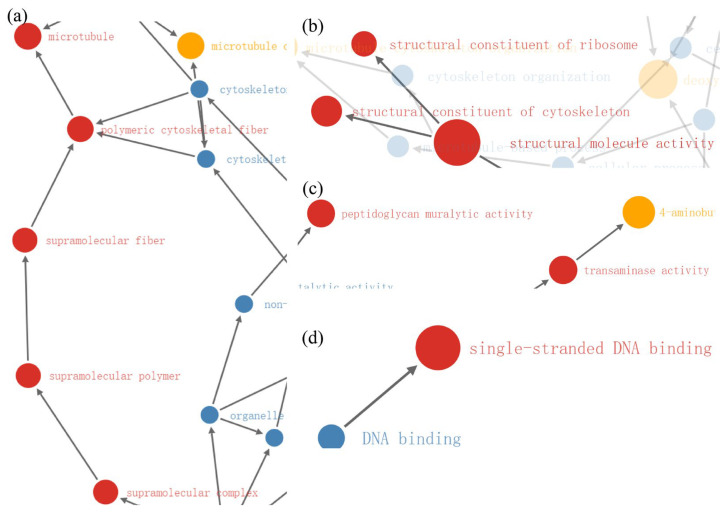
Functional enrichment diagram of 400 Gy/CK group. (**a**–**d**) Functional enrichment diagram of down-regulated proteins. Red indicates *p* < 0.01; yellow indicates *p* < 0.05; blue indicates no significance.

**Figure 15 molecules-28-05727-f015:**
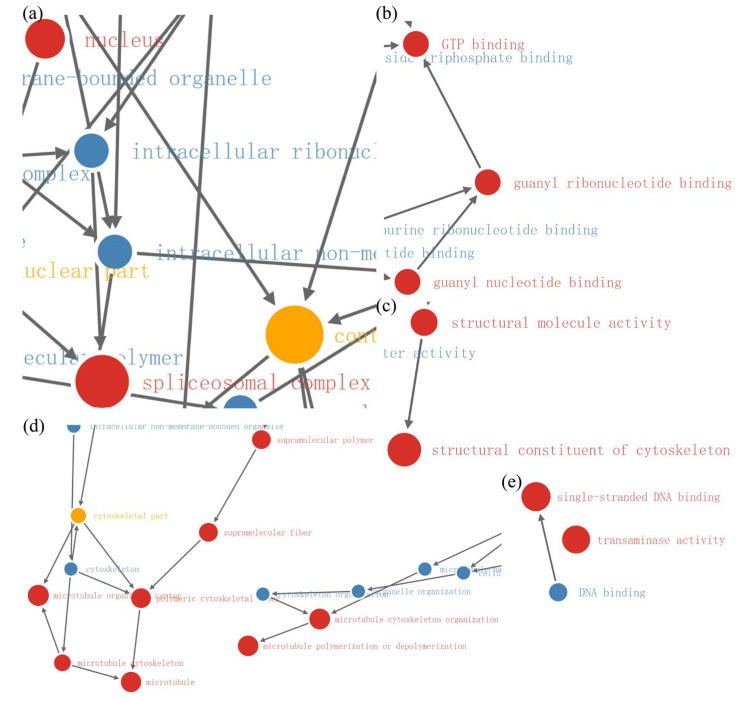
Functional enrichment diagram of 400 Gy/200 Gy group. (**a**) Functional enrichment diagram of up-regulated proteins. (**b**–**e**) Functional enrichment diagram of down-regulated proteins. Red indicates *p* < 0.01; yellow indicates *p* < 0.05; blue indicates no significance.

**Figure 16 molecules-28-05727-f016:**
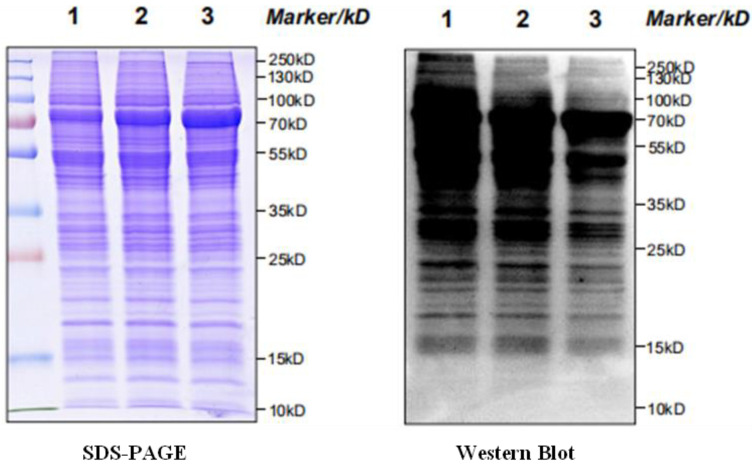
Expression characteristics of *P. xylostella* testis tyrosine phosphorylated proteins in each treatment group. Primary antibody: Anti-Phosphotyrosine Antibody. Secondary antibody: Goat Anti-Mouse IgG. (1) 0 Gy; (2) 200 Gy; (3) 400 Gy.

**Table 1 molecules-28-05727-t001:** Preparation of standard curve for protein quantification.

Substance	Serial Number					
	1	2	3	4	5	6
Bovine serum albumin standard solution (μL)	0	2	4	6	8	10
PBS buffer (μL)	20	18	16	14	12	10
Coomassie brilliant blue G-250 solution (μL)	200	200	200	200	200	200
Protein content (μg)	0	2	4	6	8	10
OD_562_	0.3705	0.4970	0.6125	0.7695	0.8580	0.9240

**Table 2 molecules-28-05727-t002:** Major bioinformatics analyses software/website.

Analyze Project	Software/ Method	Website
Mass spectrum data analysis	MaxQuant	http://www.maxquant.org/ (accessed on 3 April 2022)
Motif analysis	MoMo	http://meme-suite.org/tools/momo (accessed on 6 April 2022)
GO annotation	InterProScan	http://www.ebi.ac.uk/interpro/ (accessed on 6 April 2022)
Domain annotation	InterProScan	http://www.ebi.ac.uk/interpro/ (accessed on 6 April 2022)
KEGG annotation	KEGG Mapper	http://www.kegg.jp/kegg/mapper.html (accessed on 6 April 2022)
	KAAS	http://www.genome.jp/kaas-bin/kaas_main (accessed on 6 April 2022)
Subcellular localization	Wolfpsort	http://www.genscript.com/psort/wolf_psort.html (accessed on 6 April 2022)
	CELLO	http://cello.life.nctu.edu.tw/ (accessed on 6 April 2022)
Enrichment analysis	Perl module	https://metacpan.org/pod/Text::NSP::Measures::2D::Fisher (accessed on 8 April 2022)
Clustering heat map	R Package pheatmap	https://cran.r-project.org/web/packages/cluster/ (accessed on 8 April 2022)
Protein interaction	Blast	http://blast.ncbi.nlm.nih.gov/Blast.cgi (accessed on 8 April 2022)
	R package networkD3	https://cran.r-project.org/web/packages/networkD3/ (accessed on 8 April 2022)

## Data Availability

Not applicable.

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
