# Peer review of "Molecular Mechanism of Male Sterility Induced by 60Co γ-Rays on Plutella xylostella (Linnaeus)"

_molecules, 2023, doi:10.3390/molecules28155727_

Round 1

Reviewer 1 Report (New Reviewer)

A quantitative proteomics analysis was used for investigating the testicular tissue of P.xylostella after irradiation. The related proteomic changes (up and down regulation) in the testes of the male P.xylostella were identified by analyzing the DEPs, the differential enrichment analysis of proteomics. The authors conducted the bioinformatics analysis on the quantitative data of proteome and the data of DEPs to find the change rule of DEPs. The expected standard has been followed and all relevant analyses are presented in figures.  

Expression characteristics of P.xylostella testis tyrosine phosphorylated proteins in each treatment group have been made in western blot analysis.

After 200Gy irradiation, the testes are shown to resist radiation damage by increasing energy supply, amino acid metabolism and transport, and protein synthesis. But transcription-related pathways were inhibited. Following a 400Gy irradiation, the mitochondria and DNA in the testis tissue of P.xylostella were damaged, which caused cell autophagy and apoptosis, affected the normal life activities of sperm cells, and greatly weakened sperm motility and insemination ability. Compared with the irradiation dose of 200Gy, the irradiation dose of 400Gy promoted the expression of signal pathways related to proteolysis and gene expression in tissues and cells and inhibited the expression of signal pathways related to protein synthesis and cytoskeleton synthesis. These reports are found consistent with their presented data.

The article is generally prepared with professional norms. Good luck!

Author Response

Dear reviewer,

I am very grateful to your comments for the manuscript.

Wish you a happy life!

Reviewer 2 Report (New Reviewer)

The authors have represented a promising strategy for pest control and the results have been elaborated following a detailed methodology. However, it would be clear if the authors could bring the methodology section after introduction and separate results and discussion . Most of the figures are very small and unclear. Either higher resolution figures shall be uploaded or the size shall be made bigger. The authors shall choose a better set of keywords.

Overall, the language seems good and understandable, free of any evident grammatical errors.

Author Response

Dear reviewer,

I am very grateful to your comments for the manuscript. According with your advice, we amended the relevant part in manuscript. Some of your questions were answered below. Specific amended in the attachment.

Point 1: The authors have represented a promising strategy for pest control and the results have been elaborated following a detailed methodology. However, it would be clear if the authors could bring the methodology section after introduction and separate results and discussion . Most of the figures are very small and unclear. Either higher resolution figures shall be uploaded or the size shall be made bigger. The authors shall choose a better set of keywords.

Response 1: The authors think that the suggestion of reviewer is advisable. We have revised the discussion section and added some content in the introduction. The keyword has been changed. We uploaded a higher resolution picture.

Wish you a happy life!

Reviewer 3 Report (New Reviewer)

To,

The Editor,

Molecules, MDPI,

Manuscript ID: molecules-2489770

Subject: Submission of comments of the manuscript in “Molecules"

Dear Editor Molecules, MDPI,

Thank you very much for the invitation to consider a potential reviewer for the manuscript (ID: molecules-2489770). My comments responses are furnished below as per each reviewer’s comments. 

In the reviewed manuscript, the authors used proteomics technology and bioinformatics analysis to investigate the molecular mechanism of the effects of different doses of radiation treatment on the reproductive ability of male P. xylostella. The results showed that after 200Gy irradiation, the testes resisted radiation damage by increasing energy supply, amino acid metabolism and transport, and protein synthesis. But transcription-related pathways were inhibited. After 400Gy irradiation, the mitochondria and DNA in the testis tissue of P.xylostella were damaged, which caused cell autophagy and apoptosis, affected the normal life activities of sperm cells, and greatly weakened sperm motility and insemination ability. Compared with the irradiation dose of 200Gy, the irradiation dose of 400Gy promoted the expression of signal pathways related to proteolysis and gene expression in tissues and cells and inhibited the expression of signal pathways related to protein synthesis and cyto-skeleton synthesis. Meanwhile, Western Blot showed that irradiation had an effect on tyrosine phosphorylation level, and with the increase of irradiation dose, tyrosine phosphorylation level gradually decreased. However, in my opinion, the MS needs major revisions. I have some suggestions to improve this manuscript: 

  1. I have read the entire manuscript and my initial comment is that manuscript is poorly written. I have significant concerns about the grammar and vocabulary of the manuscript; therefore, I recommend the authors to use an English proofreading service.
  2. The abstract does not reflect the whole story, revise it
  3. The key words must be in alphabetical order.
  4. The writing style of the paper is very poor. There are many grammatical mistakes. Long sentences with noticeable grammatical mistakes are frequently present throughout the manuscript. There are many typos mistakes in this whole manuscript. The author should check the whole manuscript.
  5. The introduction part is not impressive and systematic. In the introduction part, the authors should elaborate on the scientific issues in plant research. The Content of the introduction is effective in essence but very poorly presented, significant improvements are needed in presenting the proper background of the work undertaken
  6. Figures 1, 2, 3, 4, 5, 7, 8, 9, 10, 11, 12, 13, 14, and 15 have quite low resolution and are difficult to make out. Further, figure texts are not readable. Higher-resolution versions will be needed for publication,
  7. In Material and Methods:- indicate how many replicates assayed in each analysis/parameter. The number of samples or biological and technical replicates should be mentioned for each parameter in the methods.
  8. The discussion should be interpreted with the results as well as discussed in relation to the present literature.  
  9. References: shall have to correct the whole References according to the ”Instructions for the Authors”, e.g. the Journal name must be abbreviated, the journal name in italics, the year must be bold and the author shall have to use the without italics paper titles.
  10. The conclusion is very lengthy and requires improvement.

 Best wishes and thank you for the opportunity.

To,

The Editor,

Molecules, MDPI,

Manuscript ID: molecules-2489770

Subject: Submission of comments of the manuscript in “Molecules"

Dear Editor Molecules, MDPI,

Thank you very much for the invitation to consider a potential reviewer for the manuscript (ID: molecules-2489770). My comments responses are furnished below as per each reviewer’s comments. 

In the reviewed manuscript, the authors used proteomics technology and bioinformatics analysis to investigate the molecular mechanism of the effects of different doses of radiation treatment on the reproductive ability of male P. xylostella. The results showed that after 200Gy irradiation, the testes resisted radiation damage by increasing energy supply, amino acid metabolism and transport, and protein synthesis. But transcription-related pathways were inhibited. After 400Gy irradiation, the mitochondria and DNA in the testis tissue of P.xylostella were damaged, which caused cell autophagy and apoptosis, affected the normal life activities of sperm cells, and greatly weakened sperm motility and insemination ability. Compared with the irradiation dose of 200Gy, the irradiation dose of 400Gy promoted the expression of signal pathways related to proteolysis and gene expression in tissues and cells and inhibited the expression of signal pathways related to protein synthesis and cyto-skeleton synthesis. Meanwhile, Western Blot showed that irradiation had an effect on tyrosine phosphorylation level, and with the increase of irradiation dose, tyrosine phosphorylation level gradually decreased. However, in my opinion, the MS needs major revisions. I have some suggestions to improve this manuscript: 

  1. I have read the entire manuscript and my initial comment is that manuscript is poorly written. I have significant concerns about the grammar and vocabulary of the manuscript; therefore, I recommend the authors to use an English proofreading service.
  2. The abstract does not reflect the whole story, revise it
  3. The key words must be in alphabetical order.
  4. The writing style of the paper is very poor. There are many grammatical mistakes. Long sentences with noticeable grammatical mistakes are frequently present throughout the manuscript. There are many typos mistakes in this whole manuscript. The author should check the whole manuscript.
  5. The introduction part is not impressive and systematic. In the introduction part, the authors should elaborate on the scientific issues in plant research. The Content of the introduction is effective in essence but very poorly presented, significant improvements are needed in presenting the proper background of the work undertaken
  6. Figures 1, 2, 3, 4, 5, 7, 8, 9, 10, 11, 12, 13, 14, and 15 have quite low resolution and are difficult to make out. Further, figure texts are not readable. Higher-resolution versions will be needed for publication,
  7. In Material and Methods:- indicate how many replicates assayed in each analysis/parameter. The number of samples or biological and technical replicates should be mentioned for each parameter in the methods.
  8. The discussion should be interpreted with the results as well as discussed in relation to the present literature.  
  9. References: shall have to correct the whole References according to the ”Instructions for the Authors”, e.g. the Journal name must be abbreviated, the journal name in italics, the year must be bold and the author shall have to use the without italics paper titles.
  10. The conclusion is very lengthy and requires improvement.

 Best wishes and thank you for the opportunity.

Author Response

Dear reviewer,

I am very grateful to your comments for the manuscript. According with your advice, we amended the relevant part in manuscript. Some of your questions were answered below. Specific amended in the attachment.

Point 1: I have read the entire manuscript and my initial comment is that manuscript is poorly written. I have significant concerns about the grammar and vocabulary of the manuscript; therefore, I recommend the authors to use an English proofreading service.

Response 1: We have re-proofread the whole article for English writing.

Point 2: The abstract does not reflect the whole story, revise it.

Response 2: We have revised and added to the abstract.

Point 3: The key words must be in alphabetical order.

Response 3: We re-selected the appropriate keywords and put them in order.

Point 4: The writing style of the paper is very poor. There are many grammatical mistakes. Long sentences with noticeable grammatical mistakes are frequently present throughout the manuscript. There are many typos mistakes in this whole manuscript. The author should check the whole manuscript.

Response 4: We proofread the writing of the whole article.

Point 5: The introduction part is not impressive and systematic. In the introduction part, the authors should elaborate on the scientific issues in plant research. The Content of the introduction is effective in essence but very poorly presented, significant improvements are needed in presenting the proper background of the work undertaken.

Response 5: We have revised the introduction and added some content.

Point 6: Figures 1,2,3,4,5,7,8,9,10,11,12,13,14, and 15 have quite low resolution and are difficult to make out. Further, figure texts are not readable. Higher-resolution versions will be needed for publication.

Response 6: We reuploaded the Figures in higher resolution.

Point 7: In Material and Methods:- indicate how many replicates assayed in each analysis/parameter. The number of samples or biological and technical replicates should be mentioned for each parameter in the methods.

Response 7: We have added to the material and methods the number of repetitions per treatment, the number of samples, etc.

Point 8: The discussion should be interpreted with the results as well as discussed in relation to the present literature.  

Response 8: We have revised the discussion.

Point 9: References: shall have to correct the whole References according to the ”Instructions for the Authors”, e.g. the Journal name must be abbreviated, the journal name in italics, the year must be bold and the author shall have to use the without italics paper titles.

Response 9: We have modified the format of the references in accordance with the requirements of the journal.

Point 10: The conclusion is very lengthy and requires improvement.

Response 10: We have revised the conclusion and reduced the number of words.

Wish you a happy life!

Round 2

Reviewer 2 Report (New Reviewer)

The authors revised the manuscript satisfactorily therefore it can be accepted as such 

The English text in the manuscript is readable. 

Reviewer 3 Report (New Reviewer)

Dear Chief Editor,

Thank you for providing the opportunity to review the revised manuscript. The authors have addressed all comments and incorporated changes suggested by reviewers during the first round of revisions. The revised version of the manuscript is improved as expected. Based on these revisions, now this study is a suitable contribution to the Molecules. I recommend the manuscript for publication.

Thank you

With best regards

Dear Chief Editor,

Thank you for providing the opportunity to review the revised manuscript. The authors have addressed all comments and incorporated changes suggested by reviewers during the first round of revisions. The revised version of the manuscript is improved as expected. Based on these revisions, now this study is a suitable contribution to the Molecules. I recommend the manuscript for publication.

Thank you

With best regards

This manuscript is a resubmission of an earlier submission. The following is a list of the peer review reports and author responses from that submission.

Round 1

Reviewer 1 Report

The presented article was written by bioinformaticians and is intended for bioinformaticians as well. The experimental work is based on obtaining a specific biomaterial and performing of competitive experiments using a cobalt gun. In the introduction, the authors write that sterile insect technology is promising approach, I ask the authors to explain how they plan to put the SIT into practice. Are authors suggesting to process crops in the warehouse or plants in fields? Even with luck, I don’t think that more than half of the animals will be sterilized, which will be realized in a moderate inhibition of the population growth. Radiation can damage crops and store radioactive materials or free radicals in food.

Enlargement of axis labels and text is required for all figures.

The figure 2 shows data for proteins based on the identified tryptic peptides, the % of coverage of the full sequence is not indicated in the methods, so the authors could identify the protein by a single peptide of 7 residues. Irradiation leads to random mutations and destruction/replacement of nucleotides, which in turn leads to proteins mutations and stop codon occurs, therefore the protein with multiple mutations can detected as several new proteins (with low coverage %) and be false positives. As a result, data about suppression or increased production of a particular protein are poorly substantiated. The fact that more than 100 proteins suddenly cease to be observed after banal irradiation is hard to believe.

Figure 16 shown solo experiment in biology field with antibody staining, but does not save the work from “failure”. To many standard bioinformatics charts in the results are not supported by wet biology experimental data, therefore the manuscript have low interest for broad scientific community and should be published in more specialized journals.

Reviewer 2 Report

The authors find that 0Co-γ Rays 2 on Plutella xylostella can Molecular Mechanism of Male Sterility Induced. Overall, this manuscript was good but it needs to be improved. I write some suggestion and comments below.

1-     In introduction part, “In recent years, compared with the traditional differential electrophoresis …” is not cited.

2-     Which database do authors use to find Classification of differential protein?

3-     Which software do authors use to find Venn diagram analysis, Enrichment analysis and signaling pathway?

4-     Figure 13 (Functional enrichment diagram) is low resolution.

5-     Which database do author draw Functional enrichment diagram?

6-     Line “Among them, NUP205 and NUP155 were nuclear porins, and the changes of nuclear porins in...”, the citation is wrong.

7-     Does this result support this sentence? “At the same time, it induces apoptosis in the testis through exogenous death receptor pathway and mitochondrial pathway, resulting in reproductive toxicity.”

Moderate editing of English language required